# Ultrahigh Energy Storage Properties in (Sr_0.7_Bi_0.2_)TiO_3_-Bi(Mg_0.5_Zr_0.5_)O_3_ Lead-Free Ceramics and Potential for High-Temperature Capacitors

**DOI:** 10.3390/ma13010180

**Published:** 2020-01-01

**Authors:** Xi Kong, Letao Yang, Zhenxiang Cheng, Shujun Zhang

**Affiliations:** ISEM, Australian Institute of Innovative Materials, University of Wollongong, Wollongong NSW 2500, Australia; xk759@uowmail.edu.au (X.K.); ly742@uowmail.edu.au (L.Y.); cheng@uow.edu.au (Z.C.)

**Keywords:** dielectric capacitors, lead-free, high temperature, energy storage

## Abstract

Due to the enhanced demand for numerous electrical energy storage applications, including applications at elevated temperatures, dielectric capacitors with optimized energy storage properties have attracted extensive attention. In this study, a series of lead-free strontium bismuth titanate based relaxor ferroelectric ceramics have been successfully synthesized by high temperature solid-state reaction. The ultrahigh recoverable energy storage density of 4.2 J/cm^3^ under 380 kV/cm, with the high efficiency of 88%, was obtained in the sample with *x* = 0.06. Of particular importance is that this ceramic composition exhibits excellent energy storage performance over a wide work temperature up to 150 °C, with strong fatigue endurance and fast discharge speed. All these merits demonstrate the studied ceramic system is a potential candidate for high-temperature capacitors as energy storage devices.

## 1. Introduction

Dielectric capacitors have been widely used as energy-storage components in various electronic and electrical applications [1,2], with more focus in high-power and pulse-power applications, because of their higher power density, faster charge–discharge rate, and robustness when compared with fuel cells and batteries [3]. However, the energy density of dielectric capacitors is usually lower than other energy storage devices, which limits their applications. In addition, modern dielectric capacitors are challenged by operating under high-temperature environments, for example, the power inverters used for hybrid vehicles are operated at temperatures over 140 °C [4] and the devices for exploring underground oil or gas are required to withstand temperature over 200 °C [5]. The low glass transition point of present commercial polymer capacitors limit their applications at elevated temperature [6]. In contrast, ceramic capacitors benefit from their inorganic structure, and so are more capable for hard environments. However, the energy efficiency of ceramic capacitors usually reduces as the operating temperature increases due to the leakage current and/or space charge effect [2]. The reduced energy efficiency at elevated temperature corresponds to severe energy dissipation in the form of heat, raising the concerns of device stability and reliability. Therefore, a demand exists for exploration of high temperature dielectric ceramics with good energy storage properties.

The energy density and efficiency of capacitors can be calculated by the following the Equations:(1)$$W\;=\;\int_{0}^{P_{max}}EdP,$$
(2)$$W_{rec}\;=\;\int_{P_{r}}^{P_{max}}EdP,$$
(3)$$\eta \;=\;\frac{W_{rec}}{W}\times 100\%,$$
where *W* and *W*_rec_ represent the stored energy density and recoverable energy density. The energy efficiency (*η*) is defined by the ratio between *W*_rec_ and *W*. It can be seen that high maximum polarization (*P*_max_), low remnant polarization (*P*_r_), and high breakdown strength (BDS) relating to the various electric field will greatly benefit the recoverable energy density and energy efficiency. However, the *P*_max_ or *P*_r_ for many dielectric materials are strongly temperature-dependent [7], leading to inferior stability of the dielectric capacitors. In addition, at elevated temperature, the trapped charges existing in the ceramics gain enough energy and contribute to the conductivity, resulting in an increased leakage current and decreased energy efficiency, which will accelerate thermal breakdown of the ceramics by generating more heat [8,9]. The opposite way, the capacitors with high energy efficiency could operate at elevated temperature without additional cooling devices, which reduces the volume and cost of the whole system. Many lead-containing materials attracted wide attention as energy storage capacitors in last few decades due to their excellent properties, such as PbZrO_3_ and (Pb,La)(Zr,Ti)O_3_ (PLZT)-based systems [10,11]. However, considering the environmental and health concerns of the lead component, lead-free dielectrics have been the mainstay in research over the last 20 years.

Strontium titanate (SrTiO_3_, ST) has been extensively studied for energy storage applications in the past few years due to its high permittivity (~300), low dielectric loss (<1%) and relatively high BDS (~200 kV/cm) [12]. Many efforts have been attempted for improving the energy storage properties of ST, such as modification with the Bi^3+^ cation, Ca^2+^ cation or rare earth cations for increasing permittivity [13,14,15], meanwhile enhancing the BDS [12]. Of particular importance is that the bismuth-containing relaxor ferroelectrics have attracted increasing attention as promising alternatives for lead-free relaxors, due to the fact that the Bi^3+^ ions possess the lone-pair electronic configuration 6*s*^2^ being analogous to Pb^2+^ ions, which can be hybridized with O 2*p* orbitals, resulting the high saturation polarization. It was reported that bismuth-modified strontium titanate ((Sr_0.7_Bi_0.2_)TiO_3_ abbreviated to SBT) dielectric material exhibited strong relaxor-like behavior, showing smeared phase transition temperature and high polarization [14], while the energy storage density of 1.63 J/cm^3^ at 217.6 kV/cm was reported for relaxor ceramic (Sr_0.85_Bi_0.1_)TiO_3_ [16]. In addition, SBT has been selected as an endmember to form solid solutions with other compounds to improve the dielectric and energy storage properties. It was reported that a 0.55(Na_0.5_Bi_0.5_)TiO_3_–0.45SBT multilayer ceramic capacitor showed a high energy density of 9.5 J/cm^3^ and energy efficiency of 92% at 720 kV/cm [17]. Although the maximum energy density of 48.5 J/cm^3^ was obtained in 0.9SBT–0.1BiFeO_3_ thin-film capacitor, the energy efficiency was relatively low (48%) [18].

Based on the unique relaxor characteristic of Bi-based compound, in order to improve the temperature stability and energy storage density, numerous Bi(Me′Me″)O_3_ (Me′ = Mg^2+^, Zn^2+^; Me″ = Ti^4+^, Nb^5+^, et al.) endmembers have been attempted for dielectric solid solutions. For example, BaTiO_3_–Bi(Me′Me″)O_3_ ceramics transfer from typical ferroelectric to relaxor with increasing Bi(Me′Me″)O_3_ concentration, showing improved temperature stability, low dielectric loss, as well as improved energy efficiency [9,19,20,21,22,23]. Among these solid solutions, 0.7BaTiO_3_–0.3Bi(Mg_0.5_Zr_0.5_)O_3_ was found to exhibit high stability over a wide temperature range from −20 °C to 430 °C with low permittivity variation (< ±15%) and relatively low loss (~2%) [24]. Therefore, in this work, Bi(Mg_0.5_Zr_0.5_)O_3_-modified SBT relaxor ferroelectric solid solution (1-*x*)(Sr_0.7_Bi_0.2_)TiO_3_-*x*Bi(Mg_0.5_Zr_0.5_)O_3_ ((1-*x*)SBT-*x*BMZ) was fabricated, and its energy storage properties were studied in detail. A high energy storage density of 4.2 J/cm^3^ at applied electric field of 380 kV/cm was obtained for the *x* = 0.06 sample. The energy efficiency at 200 kV/cm was maintained above 90% over the temperature range of 25 °C to 125 °C, exhibiting an excellent temperature stability.

## 2. Materials and Methods

(1-*x*)SBT-*x*BMZ ceramics with *x* = 4%, 6%, 8%, 10%, 15% (abbreviated as 4BMZ, 6BMZ, 8BMZ, 10BMZ, and 15BMZ, respectively) were fabricated by a conventional high-temperature solid state reaction method using reagent powders SrCO_3_ (99.9%, Alfa Aesar, Haverhill, MA, USA), TiO_2_ (99.0%, Strem Chemicals, Newburyport, MA, USA), Bi_2_O_3_ (99.9%, Alfa Aesar), MgO (99.0%, Alfa Aesar), and ZrO_2_ (99.7%, Alfa Aesar). These powders were baked at 150 °C for 12 h to remove the absorbed moisture and then stoichiometrically weighed with the addition of 1–2% excess MgO [25]. The mixed powders were ball milled for 12 h with ethanol as medium. Thereafter, the powders were re-milled again after calcination at 900 °C for 2 h. The median particle size of re-milled powers were around 0.5–0.7 μm. Then 0.2 wt % Rhoplex binder was added to the slurry of each composition and the mixtures were ball-milled for 2 h. The slurries with binder were dried, granulated, and sieved. Then the dried powders were pressed into 15-mm-diameter pellets. In order to increase the green density, these pellets were pressed under an isostatic pressure of 200 MPa for 10 min after burning out the binder at 600 °C. The pellets were sintered at 1200–1250 °C for 2 h with self-source powder in a closed crucible. For dielectric and electrical measurements, the silver electrodes and gold electrodes were prepared on polished parallel sample surfaces, respectively. In addition, the pellets were polished to ~100 μm for *P*-*E* and breakdown strength measurements. The X-ray diffraction patterns were recorded on a diffractometer (GBC MMA XRD) using Cu Kα radiation with the ground pellets. The microstructure information of polished and thermally etched surfaces were collected using scanning electron microscopy (SEM) (JSM-7500FA, JEOL, Akishima Tokyo). Grain size distributions were calculated using Nano Measurer by measuring the selected area of SEM images. In order to measure the dielectric properties of the studied samples, silver electrodes were fired on both sides of polished samples. The dielectric data was collected on a *LCR* meter (4980AL, Keysight, Santa Rosa, CA, USA) from 100 Hz to 1 MHz over temperature range of −100 °C to 400 °C. For the electrical measurements, various electric fields were applied to the sample at 10 Hz and room temperature using a ferroelectric tester (TF2000, aixACCT, Aachen, Germany) to obtain *P*-*E* hysteresis loops. Thermal stability and fatigue endurance were measured at 200 kV/cm from 25–150 °C and 1 to 10^5^ cycles, respectively. The charge–discharge rate was measured with charge–discharge system (PK-CRP1701, PolyK, State College, PA, USA) based on the resistance–capacitance circuit with 2 kΩ load resistance.

## 3. Results and Discussion

### 3.1. Phase Structure and Microstructure Characterization

The XRD patterns of (1−*x*)SBT-*x*BMZ ceramics with *x* = 4–15% are shown in Figure 1a. It can be seen that all samples exhibit pseudocubic perovskite structure, as evidenced by the single (111) and (200) peaks. Figure 1b shows the enlarged (200) peaks shifting to a lower angle with increasing BMZ concentration, corresponding to the expansion of unit cell volume. It can be attributed to the larger effective ionic radius (*r*) of (Mg_0.5_Zr_0.5_)^3+^ (*r* (Mg_0.5_Zr_0.5_)^3+^ = 0.72 Å calculated from the average ionic radii of Mg^2+^ and Zr^4+^) than that of Ti^4+^ (*r*(Ti^4+^) = 0.605 Å) on the B-site. The lattice parameters are obtained from XRD patterns, the linearly increasing trend with BMZ concentration from *x* = 4% to 15% can be seen in Figure 1c. An additional peak around 30° corresponding to the secondary phase Bi_2_Ti_2_O_7_ was observed for BMZ in the range of 6–8%, above which, the secondary phase was changed to Sr_2_Bi_4_Ti_5_O_18_ [25]. The densities of SBT-BMZ samples are above 95% theoretical, being 5.46 g/cm^3^ for x = 0.04, slightly increasing to 5.68 g/cm^3^ for *x* = 0.15.

The SEM images of studied ceramics with thermally etched surface are shown in Figure 2. All (1−*x*)SBT-*x*BMZ ceramics not only show minimal porosity but also have fine grains with average grain size smaller than 1 μm, as shown in Figure 2i,j. The dense microstructure with fine grains are expected to benefit the enhancement of BDS [26].

### 3.2. Dielectric and Energy-Storage Characterizations for (1−x)SBT-xBMZ Ceramics

The dielectric properties of BT-Bi(Me′Me”)O_3_-based ceramics have been widely investigated to acquire deeper understanding, including the relationship of relative dielectric permittivity (*ε*_r_) and the component Bi(Me′Me”)O_3_ endmember. Several previous studies show that the phase transition peaks became broader and smeared with increasing Bi(Me′Me”)O_3_ substitutions in BT-Bi(Me′Me”)O_3_ solid solutions, leading to the enhancement of thermal stability [23,24,27,28]. Figure 3a–e shows the temperature dependence of dielectric permittivity (*ε*_r_) and loss (tan*δ*) for (1−*x*)SBT-*x*BMZ ceramics. The smeared phase transition and frequency dispersion behavior of permittivity and dielectric loss are observed near the dielectric maximum temperature (*T*_m_), exhibiting a typical relaxor behavior. Generally, the diffuseness of phase transition can be evaluated by modified Curie–Weiss law 1/*ε* − 1/*ε*_m_ = (*T* − *T*_m_)*^γ^*/C, where *ε*_m_ and *T*_m_ represent maximum permittivity and the corresponding temperature, respectively. The value *γ* = 2 represents a complete diffuse phase transition while *γ* = 1 represents a classical ferroelectric transition [29]. In order to reduce the effects of space charge, the *γ* values for (1-*x*)SBT-*x*BMZ ceramics are calculated using permittivity data at 1 MHz. Figure 3f gives the *γ* values of (1-*x*)SBT-*x*BMZ ceramics, being above 1.5, indicating relaxor characteristics. With increasing BMZ, the maximum permittivity (*ε*_m_) at 1 kHz for ceramics with *x* up to 10% are similar, but the *T*_m_ increases gradually from −68 °C to −27 °C, higher than the *T*_m_ of (Sr_0.7_Bi_0.2_)TiO_3_ ceramics which is around −100 °C [30]. The lower permittivity and higher dielectric loss for samples with x = 15% may be related to the secondary phase. Of particular interest is that the dielectric loss is less than 3% for all the studied ceramics over a wide temperature range, as shown in Table 1, exhibiting the potential of high energy efficiency in the same temperature range. In addition, dielectric loss corresponds the reliability of capacitors, lower generated heat leading to the associated low possibility of thermal breakdown. It should be noted that an additional dielectric peak around 290 °C was observed for 10BMZ and 15BMZ, as shown in Figure 3d,e. This observation can be explained by the secondary phases (Sr_2_Bi_4_Ti_4_O_15_) which was reported to possess a Curie temperature around 285 °C [31].

The *P*–*E* loops at 200 kV/cm for (1−*x*)SBT-*x*BMZ ceramics with different BMZ content are compared and shown in Figure 4a. It is clear to see that the 4BMZ and 6BMZ ceramics exhibit slim *P*–*E* loops at 200 kV/cm, indicating high energy efficiencies above 92%. With increasing BMZ concentration, *P*_max_ increases first then decreases, relevant to the dielectric permittivity, as listed in Table 1. The highest *P*_max_ was found for 8BMZ, but its large *P*_r_ would reduce the recoverable energy density and decreases energy efficiency. The hysteresis increases obviously in 8BMZ, 10BMZ and 15BMZ samples, resulting in the decreased *W*_rec_ with increasing BMZ concentration, as shown in Figure 4b. The increased hysteresis may be related to the increased secondary phases. Based on the preliminary results, we chose the 6BMZ sample for further evaluation of the energy storage property.

### 3.3. Energy Storage Properties of 0.94SBT-0.06BMZ Ceramic

The unipolar *P*–*E* loops of 6BMZ ceramic are measured at 10 Hz with different applied electric fields as shown in Figure 5a. It is clear that all *P*–*E* loops are slim with negligible hysteresis which are associated with the small value of *P*_r_, contributing to the high energy efficiency. *P*_r_ and *P*_max_ as a function of electric field for 6BMZ ceramic are shown in Figure 5b. When the electric field increased from 50 to 380 kV/cm, the *P*_max_ value enhances monotonically from 6.7 μC/cm^2^ to 30.3 μC/cm^2^, while the *P*_r_ remains at almost the same values. Thus, the Δ*P* (Δ*P* = *P*_max_ − *P*_r_) keeps increasing as function of applied field, which is beneficial for enlarging *W*_rec_ and *η*; the results are given in Figure 5c. The maximum *W*_rec_ of 4.2 J/cm^3^ was obtained at 380 kV/cm. On the other hand, the energy efficiency exhibits remarkable stability over the applied field up to 320 kV/cm with values larger than 90%, above which the *η* slightly drops to 88%.

High BDS is advantageous in dielectric materials for obtaining high *W* as shown in Equation (1). The Weibull distribution has been used as a capable analysis method to explore the breakdown strength of ceramics, as given in the following equations:(4)$$X_{i}\;=\;\ln\left(E_{i}\right),$$
(5)$$Y_{i}\;=\;\ln\left(\ln\left(\frac{1}{1-P_{i}}\right)\right),$$
(6)$$P_{i\;}=\;\frac{i}{n+1}$$
where *i* is the serial number of sample. *E*_i_, *P*_i_ and *n* are the breakdown strength of each test sample, the probability of breakdown and the summation samples, respectively. The ascending sort of breakdown strength for samples is shown as $E_{1}\leq E_{2}\leq \ldots E_{i}\ldots \leq E_{n}$. The Weibull modulus (*m*) can be obtained by the slope of the fitted linear line of *X*_i_ and *Y*_i_, which is one of the important parameters that represents the reliability of experimental data. The Weibull characteristic breakdown strength (*E*_b_) can be calculated by the intercept between the fitted linear line and *Y*_i_ = 0. As shown in Figure 5d, the *m* value for 6BMZ ceramic is 17, leading to a high level of confidence. The calculated *E*_b_ of 6BMZ is 460 kV/cm which is much higher than other lead-free energy storage ceramics [32,33,34,35,36], as shown in Table 2. The energy storage properties of BaTiO_3_-Bi(Me′Me″)O_3_ ceramics have been widely studied, and the obtained energy densities have been found to be around 1–3 J/cm^3^ in previous research as shown in Table 2. The dielectric maxima temperatures of most of the studied BaTiO_3_-Bi(Me′Me″)O_3_ ceramics are around or above room temperature, leading to a relatively high dielectric loss at room temperature. However, in this work, the *T*_m_ of 6BMZ is far below room temperature, resulting in a low dielectric loss at room temperature and above. The depressed tan*δ* is believed to be associated with high breakdown strength, and hence the high energy density as well as the high energy efficiency. The ultrahigh breakdown strength of 6BMZ ceramic leads to improved *W*_rec_ of 4.2 J/cm^3^, which is favorable for energy storage applications.

For high-temperature energy-storage applications, thermal stability plays a key role when working at elevated temperatures. For example, the operating temperature can be higher than 140 °C in hybrid electric vehicles, thus requiring some electronic components to be protected by cooling systems [37]. Dielectric materials that work and maintain stability at high temperature could not only reduce the cost but also the volume of the whole energy storage system. The temperature-dependent unipolar *P*–*E* loops for 6BMZ ceramic at various temperatures from ambient to 150 °C at 200 kV/cm are shown in Figure 6a. It can be seen that all the *P*–*E* loops are slim, indicating the high temperature stability. The *P*_max_ value was found to slightly decrease, which is related to the decreased permittivity at elevated temperature, while the *P*_r_ remains at a low value over a wide temperature range, indicating the 6BMZ ceramic capable of operating at high temperature. Figure 6b shows *W*_rec_ and *η* as functions of temperature, the decrease of *W*_rec_ is on the order of 14% due to reduced permittivity at elevated temperature. Of particular significance is that the variation of *η* is minimal up to 130 °C, associated with the low dielectric loss and temperature-insensitive *P*_r_. The excellent thermal stability of *η* for 6BMZ ceramic make it very promising for energy storage applications working at elevated temperature.

### 3.4. Fatigue Endurance and Charge–Discharge Rate for 0.94SBT-0.06BMZ Ceramic

To further investigate the energy storage properties for 6BMZ ceramic, the fatigue cycling endurance and a charge-discharge rate experiment were performed. Strong fatigue endurance is essential for energy storage capacitors in practical applications. Figure 7a displays the unipolar *P*–*E* loops for 6BMZ ceramic at 200 kV/cm under different cycle numbers. The energy storage performances of the 6BMZ ceramic are reproducible after 10^5^ cycles, where the unipolar *P*–*E* loops maintain the same shape over the cycling, small difference can be seen in the inset figure of Figure 7a. The variation of *W*_rec_ and *η* for fatigue cycling numbers are given in Figure 7b; here the *W*_rec_ value remains almost the same value while the *η* slightly drops from 95% to 92% after 10^5^ cycles, demonstrating that 6BMZ sample possesses excellent cycling reliability.

The fast discharge rate is critical for pulse-power applications. The discharged energy density can be calculated by $W_{dis}\;=\;\int R_{L}i^{2}\left(t\right)dt/Vol$, where *R*_L_, *i*(*t*) and *Vol* represent the total load resistor (2 kΩ), the discharge current through *R*_L_, and the sample volume, respectively. The calculated *W*_dis_ at an applied electric field of 200 kV/cm as a function of time for 6BMZ ceramic is given in Figure 8. The discharge time *τ*_0.9_ is the time required to release 90% of total *W*_dis_ value. The *τ*_0.9_ for 6BMZ at room temperature is 0.62 μs, indicating the great potential of 6BMZ for pulse power capacitors.

## 4. Conclusions

In summary, lead-free (1−*x*)SBT-*x*BMZ ceramics were fabricated by conventional solid state processing. The highest recoverable energy density of 4.2 J/cm^3^ and high energy efficiency of 88% were obtained at applied electric field of 380 kV/cm for 0.94(Sr_0.7_Bi_0.2_)TiO_3_-0.06Bi(Mg_0.5_Zr_0.5_)O_3_ ceramic. The 6BMZ ceramic shows a good temperature stability over a wide temperature range from room temperature to 150 °C. In addition, the 6BMZ ceramic also exhibits excellent cycling stability over 10^5^ charging–discharging cycles. Together with its fast discharge rate, the τ_0.9_ is on the order of 0.62 μs, demonstrating that the 0.94(Sr_0.7_Bi_0.2_)TiO_3_-0.06Bi(Mg_0.5_Zr_0.5_)O_3_ ceramic shows promising potential as a lead-free dielectric for high-temperature, high-power energy storage applications.

## Figures and Tables

**Figure 1 materials-13-00180-f001:**
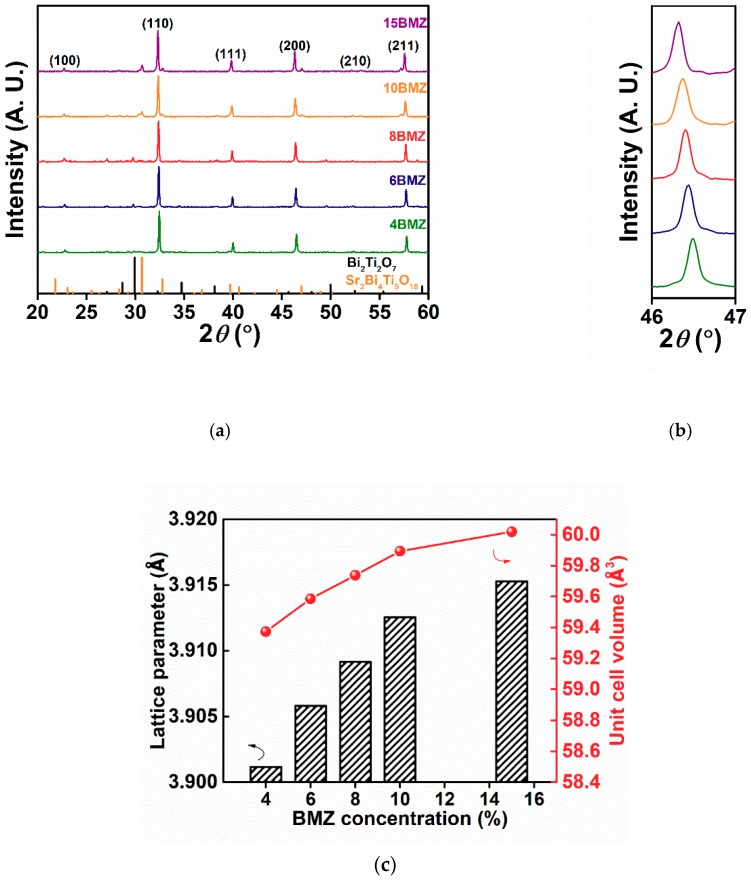
(**a**) XRD patterns of (1−*x*)(Sr_0.7_Bi_0.2_)TiO_3_-*x*Bi(Mg_0.5_Zr_0.5_)O_3_ ceramics; (**b**) enlarged diffraction peaks from 46° to 47° of (1−*x*)(Sr_0.7_Bi_0.2_)TiO_3_-*x*Bi(Mg_0.5_Zr_0.5_)O_3_ ceramics; (**c**) the lattice parameters as a function of BMZ concentration.

**Figure 2 materials-13-00180-f002:**
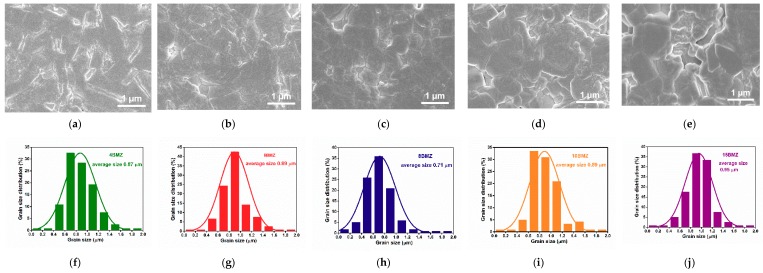
SEM micrographs of the polished and thermally etched surfaces for (1−*x*)(Sr_0.7_Bi_0.2_)TiO_3_-*x*Bi(Mg_0.5_Zr_0.5_)O_3_ ceramics: (**a**) *x* = 4%; (**b**) *x* = 6%; (**c**) *x* = 8%; (**d**) *x* = 10%; (**e**) *x* = 15%; The grain size of (1−*x*)(Sr_0.7_Bi_0.2_)TiO_3_-*x*Bi(Mg_0.5_Zr_0.5_)O_3_ ceramics: (**f**) *x* = 4%; (**g**) *x* = 6%; (**h**) *x* = 8%; (**i**) *x* = 10%; (**j**) *x* = 15%.

**Figure 3 materials-13-00180-f003:**
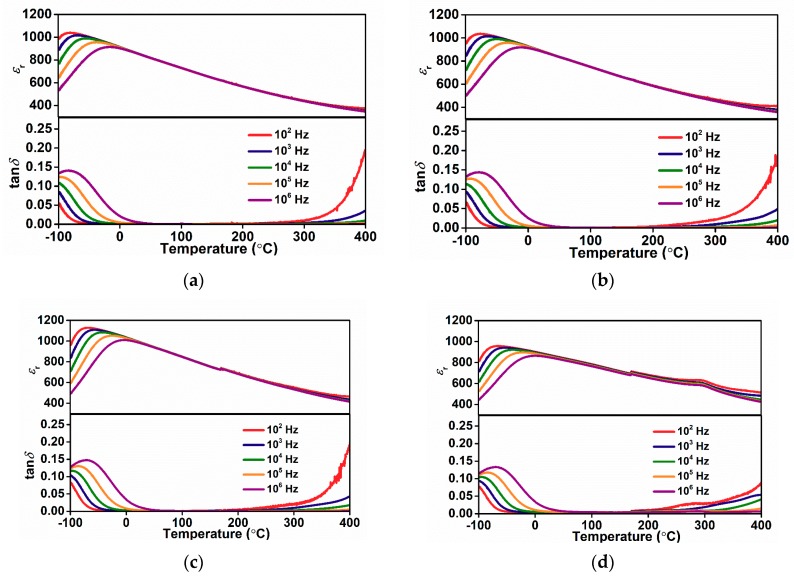
Temperature dependence of *ε*_r_ and tan*δ* of (1−*x*)(Sr_0.7_Bi_0.2_)TiO_3_-*x*Bi(Mg_0.5_Zr_0.5_)O_3_ ceramics: (**a**) *x* = 4%; (**b**) *x* = 6%; (**c**) *x* = 8%; (**d**) *x* = 10%; (**e**) *x* = 15%; (**f**) ln (1/*ε* − 1/*ε*_m_) as a function of ln (*T* − *T*_m_) for (1−*x*)(Sr_0.7_Bi_0.2_)TiO_3_-*x*Bi(Mg_0.5_Zr_0.5_)O_3_ ceramics at 1 MHz.

**Figure 4 materials-13-00180-f004:**
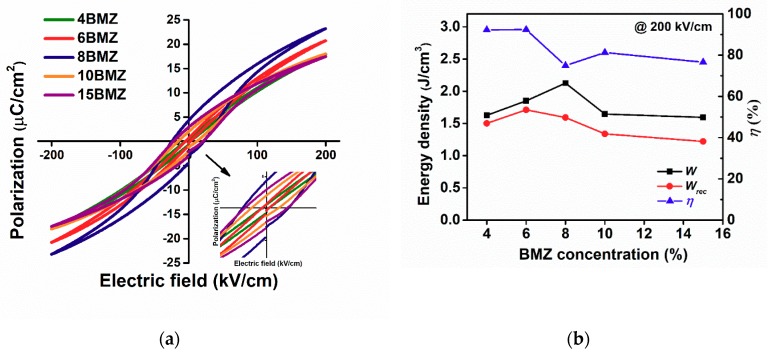
(**a**) The *P*–*E* loops at 200 kV/cm for (1−*x*)(Sr_0.7_Bi_0.2_)TiO_3_-*x*Bi(Mg_0.5_Zr_0.5_)O_3_ ceramics; (**b**) the stored energy density (*W*), recoverable energy density (*W*_r_), and energy efficiency (*η*) for (1−*x*)(Sr_0.7_Bi_0.2_)TiO_3_-*x*Bi(Mg_0.5_Zr_0.5_)O_3_ ceramics as functions of different BMZ concentration at 200 kV/cm and ambient temperature.

**Figure 5 materials-13-00180-f005:**
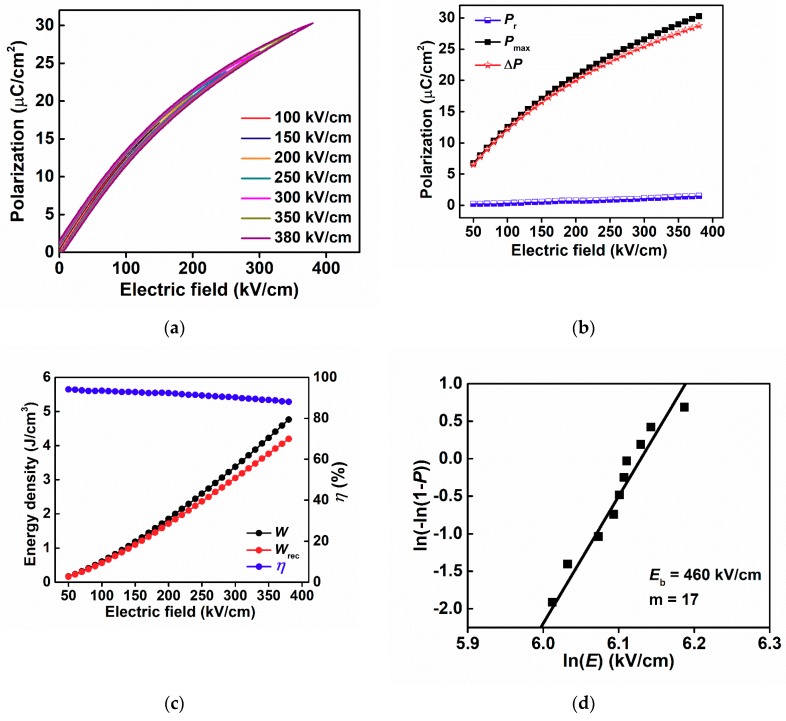
(**a**) The unipolar *P*–*E* loops for 6BMZ ceramic; (**b**) the electric filed dependent *P*_max_, *P*_r_, and Δ*P* for 6BMZ ceramic; (**c**) the calculated *W*, *W*_rec_, and *η* versus applied electric field of 6BMZ ceramic; (**d**) Weibull distribution of the breakdown strength data.

**Figure 6 materials-13-00180-f006:**
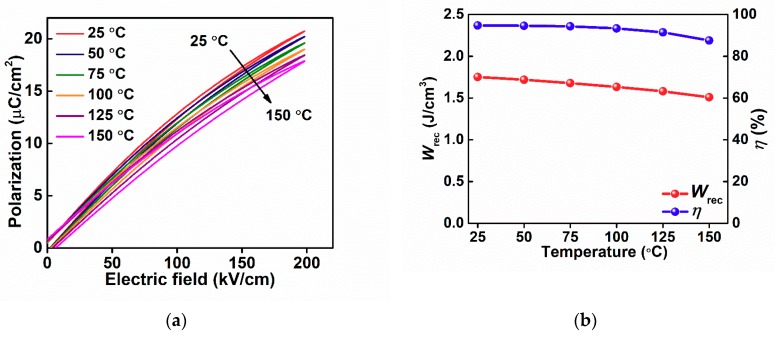
(**a**) The unipolar *P*–*E* loops for 6BMZ ceramic at different temperature at 200 kV/cm; (**b**) *W*_r_ and *η* of 6BMZ ceramic as functions of temperature.

**Figure 7 materials-13-00180-f007:**
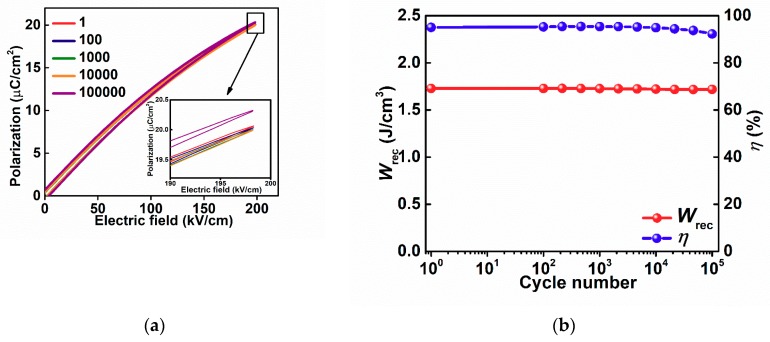
(**a**) Unipolar *P*–*E* loops for 6BMZ ceramic under different fatigue cycling number; (**b**) *W_r_* and *η* as functions of charging–discharging cycles from 1 to 10^5^ for 6BMZ ceramic.

**Figure 8 materials-13-00180-f008:**
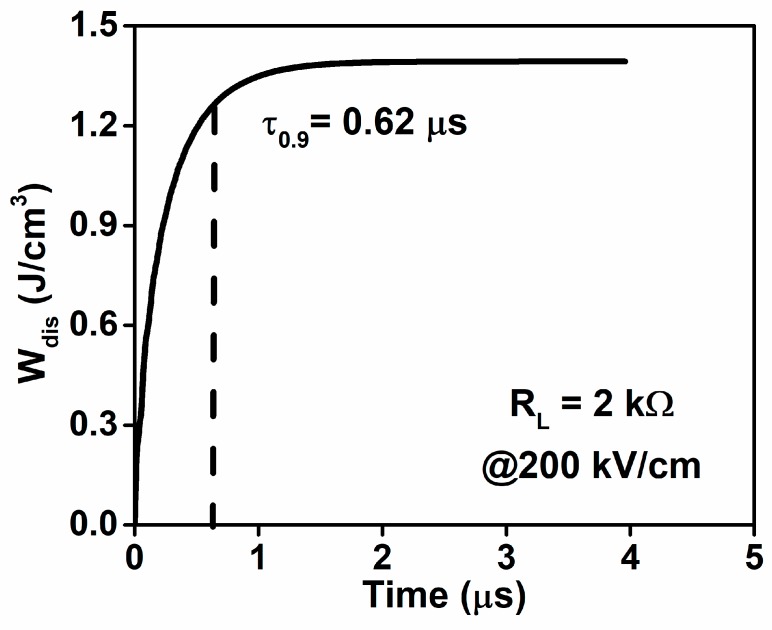
Discharge energy density as a function of time for 6BMZ ceramic at 200 kV/cm and ambient temperature.

**Table 1 materials-13-00180-t001:** The summary of dielectric performance at 1 kHz and energy storage properties at 200 kV/cm for (1−*x*)(Sr_0.7_Bi_0.2_)TiO_3_-*x*Bi(Mg_0.5_Zr_0.5_)O_3_ ceramics.

Sample	*T*_m_ (°C)	*ε* _m_	*ε* @ RT	tan*δ* @ RT	*T*-range (°C) tan*δ* < 3%	*W*_rec_ @ RT (J/cm^3^)	*η* @ RT (%)
4BMZ	−68	1020	865	0.06%	−80–400	1.50	92
6BMZ	−64	1020	880	0.06%	−75–380	1.71	92
8BMZ	−56	1110	990	0.08%	−70–390	1.59	75
10BMZ	−54	940	870	0.04%	−70–350	1.34	81
15BMZ	−27	890	865	0.2%	−40–340	1.22	77

**Table 2 materials-13-00180-t002:** Comparison of energy-storage properties between the 0.94SBT-0.06BMZ ceramic and other lead-free ceramics.

Composition	*W*_rec_ (J/cm^3^)	*η* (%)	*E*_app_ (kV/cm)	*E*_b_ (kV/cm)	Ref.
0.85BaTiO_3_-0.15Bi(Zn_2/3_Nb_1/3_)O_3_	0.79	93.5	131	262	[22]
0.9BaTiO_3_-0.1Bi(Mg_2/3_Nb_1/3_)O_3_	1.13	95.8	143.5	~270	[32]
0.61BiFeO_3_-0.33BaTiO_3_-0.06Ba(Mg_1/3_Nb_2/3_)O_3_	1.56	75	125	-	[33]
0.88BaTiO_3_-0.12Bi(Mg_1/2_Ti_1/2_)O_3_	1.81	~88	224	535.5	[20]
0.88BaTiO_3_-0.12Bi(Li_0.5_Nb_0.5_)O_3_	2.032	88	270	340	[9]
0.85BaTiO_3_-0.15Bi(Zn_0.5_Sn_0.5_)O_3_	2.21	91.6	230	280	[34]
0.9BaTiO_3_-0.1Bi(Zn_0.5_Zr_0.5_)O_3_	2.46	~73	264	266.5	[35]
(Na_0.25_Bi_0.25_Sr_0.5_)(Ti_0.8_Sn_0.2_)O_3_	3.4	90	310	330	[36]
0.94(Sr_0.7_Bi_0.2_)TiO_3_-0.06Bi(Mg_0.5_Zr_0.5_)O_3_	4.2	88	380	460	This work

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
