# Peer review of "Ultrahigh Energy Storage Properties in (Sr0.7Bi0.2)TiO3-Bi(Mg0.5Zr0.5)O3 Lead-Free Ceramics and Potential for High-Temperature Capacitors"

_materials, 2020, doi:10.3390/ma13010180_

Round 1

Reviewer 1 Report

High temperature dielectrics are an interesting topic, and this work makes a valuable contribution to the field.

However, the context of the work is poorly presented and the quality of the language and communication of information and its significance are very weak.

Authors must thoroughly revise the manuscript with the assistance of a native English speaker who has familiarity with the scientific field being discussed.

The significance of the results need to be made clear.  A presentation of the permittivity from experimental results and theoretical predictions is important.

How do these results relate to previously reported information from various other studies across the literature?? A comprehensive review and analysis of similar work is required in order to interpret the significance of the findings presented here.

What applications are these results relevant to? What was the rationale for selecting materials and processes used here?

To summarize, the work is good, the presentation is very poor.

Author Response

The authors wish to thank the reviewers for their time and efforts in providing their valuable reviews, and for constructive comments and suggestions that led to a much-improved manuscript quality. We made careful revisions based on the comments, also the point-by-point responses.

Reviewer 2 Report

This is a well crafted manuscript with interest to people working in the emerging field of energy storage in high power system. The results in terms of performance are impressive and I can anticipate that the work will have impact in the field concerned. Therefore I am recommending its publication.

Author Response

We thank the reviewer for the positive comments.

Reviewer 3 Report

In solid-state reaction method, particle size plays an important role. What was the particle size after ball grinding?

what was the relationship between the primary and secondary crystalline phases?   figure 4b. values for sample BMZ8 differ from other values. Explain these differences.

figure 5a. 6a and 7a. It is almost impossible to distinguish curves from each other.

figure 6d. It is necessary to give the relative error obtained in the calculation of BDS. These values have significant deviations from the straight line.

Author Response

(The authors gave the same response as above.)
